# Effectiveness of Humidified High Flow Nasal Cannula Versus Continuous Nasal Positive Airway Pressure in Managing Respiratory Failure in Preterm Infants: An Emergency Department Study

**DOI:** 10.3390/biomedicines13030602

**Published:** 2025-03-01

**Authors:** Duaa Yousof Mahboob, Amber Hassan, Faiza Naheed, Arshad Ali Shah, Maria Fareed Siddiqui

**Affiliations:** 1Department of Emergency, King Abdulaziz University, Jeddah 21589, Saudi Arabia; dmahbob@kau.edu.sa; 2European School of Molecular Medicine, University of Milan, 20139 Milan, Italy; 3Laboratory of Translational Neuroscience, Ceinge Biotecnologie Avanzate, 80145 Naples, Italy; 4Faculty of Allied Health Science, The University of Lahore, Lahore 54000, Pakistan; 5Faculty of Pharmacy, University of Lahore, Lahore 54000, Pakistan; faiza.naheed@pharm.uol.edu.pk; 6Department of Pharmaceutical Chemistry, Faculty of Pharmacy, University of Lahore, Lahore 54000, Pakistan; arshad.ali1@pharm.uol.edu.pk

**Keywords:** Continuous Nasal Positive Airway Pressure (nCPAP), Humidified High Flow Nasal (HHFNC), pneumonia, respiratory failure

## Abstract

**Background:** The HHFNC is routinely utilised as a non-invasive respiratory support for preterm infants with respiratory distress; few studies have compared it to nCPAP for the first treatment of respiratory distress in preterm neonates. This study aims to compare the effectiveness and outcomes of HHFNC and nCPAP in improving respiratory outcomes and reducing adverse effects. **Methods:** The 220 patients from the neonatal unit enrolled in the study (110 in each group) after obtaining written informed consent from their parents/guardians. Nasal CPAP was applied to patients in group A through a nasal mask with the following settings: FiO_2_: 40–60%, PEEP: 5–8 cm H_2_O, flow: 4–6 L/min. HHFNC was initiated at 5 L/min and adjusted between 3–7 L/min based on respiratory status, with FiO_2_ starting at 0.4 and modified to maintain SPO_2_; between 88–94%. Study variables were recorded and analysed using SPSS version 23.0. **Results:** The comparison of nCPAP (Group A) and HHFNC (Group B) showed no significant differences in age, gestational age, or clinical parameters, except for a higher respiratory rate in HHFNC. The HHFNC group had significantly shorter durations of non-invasive ventilator support and hospital stay. Adverse effects were more common in HHFNC, especially nasal mucosal injury, while sepsis was more frequent in nCPAP. Treatment failure occurred more often in the HHFNC group. Neonatal outcomes were similar, with no significant differences in discharge without the need for intubation rates, mortality, or intubation rates. **Conclusions:** The HHFNC is associated with a shorter duration of non-invasive ventilatory support and hospital stay compared to nCPAP. However, nCPAP demonstrated a significant survival advantage and a lower risk of treatment failure. Both modalities are effective in supporting preterm neonates with respiratory distress, but clinical considerations should guide the choice of therapy. Further research is necessary to confirm these findings and explore strategies to optimize outcomes and mitigate adverse effects associated with each modality.

## 1. Introduction

Respiratory distress syndrome (RDS) is a prevalent condition among preterm infants [1]. The use of non-invasive respiratory support (NRS) has been shown to lower the need for mechanical ventilation in these newborns [2]. Nasal continuous positive airway pressure (nCPAP) and heated humidified high-flow nasal cannula (HHHFNC) are widely used NRS strategies in neonatal intensive care units, offering comparable efficacy. However, their influence on feeding intolerance remains uncertain [3].

The nCPAP is the most popular non-invasive breathing assistance for preterm newborns, but it requires competent nursing care because of the bulky interfaces [4]. The HHFNC is a gentler option that is becoming popular among newborns and children with respiratory diseases [5].

The HHFNC are tiny, thin binasal prongs that supply oxygen or oxygen/air mixes at flow rates greater than 1 Litre per minute (L/min). They are located inside the nares and can take up to half of the available space without a seal [6]. Patients may tolerate larger flow rates since the gas is appropriately warmed and humidified when given. These high flow rates can equal or surpass patients’ inspiratory flow rates, limiting room air intake while ensuring that the percentage of oxygen (FiO_2_) inspired by patients matches the FiO_2_ given by the HHFNC system. The HHFNC approach is increasingly being employed for non-invasive respiratory support in preterm newborns, critically sick children, and adults [7].

The use of HHFNC in neonatal intensive care units (NI-CUs) in developed countries has significantly increased over the past six years [8]. This is owing to its effectiveness, safety, and perceived advantages over nCPAP, such as a simpler, smaller nasal interface and more comfort for babies [9,10]. Literature supports that HHFNC can avoid extubation failure, improve respiratory function, and aid in weaning off nCPAP [11]. Another study found that nasal cannulae flows of up to 2.5 L/min are equally effective as nasal nCPAP for treating apnoea of prematurity without raising oxygen requirements [12].

The studies have thoroughly examined HHFNC and nCPAP for the therapy of respiratory failure in preterm infants, emphasising their efficacy and related consequences [13,14]. This study aims to compare both treatment modalities, focusing on the mean duration of hospital stay and the mean duration of dependency on respiratory support, the ease of administration, improved compliance, and better patient outcomes compared to traditional ventilation methods.

## 2. Materials and Methods

### 2.1. Study Design and Patient Enrolment

The quasi-experimental study Reference No: (247-23) was conducted after ethical approval in 2023 from the Research Ethics Committee (REC) (IRB-SPRM/2023-24-I) in the University of Lahore Teaching Hospital, Lahore, Pakistan, and consent was taken from the parent/guardian. Patients were enrolled from the Neonatal Unit at the University of Lahore Teaching Hospital, Lahore, Pakistan. Non-probability, consecutive sampling was used to add 220 (110 in each group) patients to the study. The sample size, calculated by using the power of the study equal to 80% and a level of significance equal to 5%, taking treatment failure in the HHFNC group as 38.1% and in nCPAP as 20.9%, was 220 (110 in each group) patients [15].

### 2.2. Patient Selection and Grouping

All preterm neonates born between 32 and 34 weeks of gestation and aged 1 to 28 days, who experienced type 1 respiratory failure and needed non-invasive ventilator support, such as HHFNC or nCPAP, were enrolled in the study. We excluded patients with congenital heart disease or neuromuscular abnormalities, as well as newborns who experienced type 2 respiratory failure or required invasive mechanical ventilation or intubation.

### 2.3. Training Session

The Training session was arranged by the principal researcher for fellow doctors and nurses of the neonatal unit in which they were given detailed training regarding data collection, procedures, and interventions used in this study.

### 2.4. Data Collection

Written and informed consent was taken from parents/legal guardians for each patient after explaining the procedure and complications. The patient’s history, including gestational age, was noted. Diagnosis of respiratory failure was established with the help of clinical findings, radiological findings, and arterial blood gas analysis. Baseline variables pH, PaCO_2_, and PaO_2_ were noted by arterial blood gases. The respiratory rate was recorded. The nCPAP was applied to group 1 through a nasal mask with the following settings: FiO_2_: 40–60%, PEEP: 5–8 cm H_2_O, Flow: 4–6 L/min. HHFNC was applied to group 2 through the nasal cannula with the following settings: FiO_2_: 40–60%, Flow: 4–6 L/min. Infants receiving HHFNC therapy were initially started on a flow rate of 5 L/min, which was then adjusted between 3 and 7 L/min based on their respiratory status to maintain blood gas parameters within normal ranges. The fraction of inspired oxygen (FiO_2_) was set at 0.4 at the start of therapy and subsequently modified to ensure that oxygen saturation (SpO_2_) remained within the target range of 88–94%.

Data was recorded about the duration of hospital stay in days & duration of application of respiratory support, i.e., nCPAP or HHFNC at the time of discharge. Arterial Blood Gases were performed daily or as per the requirement to check the PO_2_ and PCO_2_. Vitals were recorded on an hourly basis, and any change of nCPAP & HHFNC settings were recorded. The same Antibiotics [Cefotaxime 150 mg/kg/24 h (Bosch Pharmaceuticals (Pvt) Ltd.; Karachi; Pakistan) & Ampicillin 200 mg/kg/24 h (Bosch Pharmaceuticals (Pvt) Ltd.; Karachi; Pakistan) as 1st line] and other treatments, including maintenance fluid, were given to both groups as per requirement. All the data was recorded on a pre-designed proforma by the staff members and results were subjected to statistical analysis by the principal researcher to determine the significance of observed differences. The data collection continued until the enrolled neonates were discharged from the hospital.

The data collection aimed to analyse the following outcomes; total duration of hospital stay in days, mean duration of application of respiratory support in hours, frequency of adverse events including sepsis (identified based on clinical symptoms, positive blood culture, and elevated inflammatory markers such as CRP), frequency of nasal mucosal injury, abdominal distension, feeding intolerance, outcome (discharged, expired, intubated and treatment failure) and neonatal outcomes (discharge, expired, intubated).

### 2.5. Data Analysis

Collected data was entered and analysed using Statistical Package for Social Science (SPSS) version 23.0. All the quantitative variables like age, APGAR score, gestational age of mother, duration of hospital, respiratory support, and total days of oxygen use up to the time of discharge for each group were reported by Mean ± SD. All the qualitative variables like gender, neonatal outcomes (discharge, expired, intubated), adverse events like sepsis, abdominal distention, feeding intolerance, and nasal mucosal injury) and outcome (discharged, expired, intubated, and treatment failure) have been presented by frequency/percentages. Duration of hospital stay in days and mean duration of application of respiratory support in hours have been compared using the Mann–Whitney U test according to normality of data. The comparison of adverse effects among both groups was compared using the Chi-square test. A *p*-value less than or equal to 0.05 was considered significant.

## 3. Results

### 3.1. Baseline Characteristics of Study Sample

The comparison of characteristics between Group A and Group B reveals no significant differences in several variables. The mean age (14.72 ± 8.45 days for Group A vs. 15.07 ± 8.02 days for Group B) and gestational age (32.71 ± 2.52 weeks for Group A vs. 32.45 ± 2.94 weeks for Group B) were similar between the two groups, with *p*-values of 0.796 and 0.472, respectively, indicating no significant differences. The gender distribution showed no notable difference (*p* = 0.495), with a slightly higher percentage of females in both groups. The Clinical parameters, including APGAR score, SPO_2_, heart rate (HR), arterial pH, PO_2_, and PCO_2_, were similar between the groups (*p* > 0.05). However, the respiratory rate was higher in Group B (55.75 ± 11.66) compared to Group A (52.29 ± 11.23) (*p* < 0.05). (Table 1).

### 3.2. Comparison of Duration of Application of Non-Invasive Ventilatory Support (Hours) and Hospital Stay Between the Groups

The comparison of the duration of application of non-invasive ventilatory support and hospital stay between the two groups shows significant differences. The mean duration of non-invasive ventilatory support was significantly shorter in Group B (HHFNC) at 65.20 h (±15.9) compared to Group A (nCPAP), which had a mean of 72.29 h (±20.7), with a *p*-value of 0.013, indicating statistical significance. Similarly, the mean duration of hospital stay was significantly shorter in Group B (HHFNC) at 20.14 days (±3.50) compared to Group A (nCPAP), which had a mean of 24.25 days (±6.07), with a *p*-value of 0.000, confirming a significant difference. These findings suggest support that HHFNC may be associated with a shorter duration of both non-invasive ventilatory support and hospital stay compared to nCPAP (Table 2). The most prevalent adverse impact in the nCPAP group was sepsis, which was documented in eight instances, followed by nasal mucosal damage in seven. Feeding intolerance and stomach distention were reported in five and four patients, respectively. In contrast, the HHFNC group exhibited a greater rate of nasal mucosal damage (18 instances) compared to nCPAP. Feeding intolerance and abdominal distention were recorded in 9 and 6 instances, respectively, whereas sepsis was diagnosed in 10 cases. These findings show that, while both groups suffered similar adverse effects, HHFNC was linked with a greater frequency of nasal mucosal damage, whereas sepsis was more common in the nCPAP group (Figure 1).

### 3.3. Comparison of Treatment Failure in Both Groups

In the study, the incidence of treatment failure was observed in both the nCPAP and HHFNC groups. Specifically, 10 patients experienced treatment failure in the nCPAP group, while 16 patients faced treatment failure in the HHFNC group. This indicates that a higher proportion of patients in the HHFNC group experienced treatment failure compared to those in the nCPAP group (Figure 2).

### 3.4. Comparison of Duration of Neonatal Outcome Between the Groups

The neonatal outcomes for both nCPAP and HHFNC groups were compared. Among the total of 220 neonates, 194 (88.2%) were discharged without the need for intubation, with 100 (90.9%) in the nCPAP group and 94 (85.5%) in the HHFNC group. The difference in discharge rates between the two groups was not statistically significant, with a *p*-value of 0.224. In terms of mortality, 9 neonates (4.1%) expired, with 5 (4.5%) from the nCPAP group and 4 (3.6%) from the HHFNC group. The causes of death included severe respiratory distress, sepsis, and complications related to prematurity. Regarding intubation, 17 neonates (7.7%) required intubation, with a higher incidence in the HHFNC group (12 cases, 10.9%) compared to the nCPAP group (5 cases, 4.5%). However, the *p*-value suggests that these differences were not statistically significant (Table 3).

The mean survival time was 30.58 days (95% CI: 29.34–31.82) for nCPAP and 22.76 days (95% CI: 21.99–23.53) for HHFNC, indicating longer survival with nCPAP. The median survival time was 33.00 days (95% CI: 30.98–35.02) for nCPAP and 24.00 days (95% CI: 21.91–26.09) for HHFNC. The Kaplan–Meier survival analysis revealed a significant difference in survival between the nCPAP and HHFNC groups (Log-Rank test: χ^2^ = 32.220, df = 1, *p* < 0.001) (Figure 3).

The nCPAP significantly reduced the risk of mortality or treatment failure compared to HHFNC (HR = 0.217, *p* < 0.001), indicating a strong survival advantage. Sepsis independently increased the risk (HR = 0.493, *p* = 0.030), highlighting its critical impact on neonatal outcomes. Treatment failure was not a significant predictor (HR = 1.128, *p* = 0.712) (Table 4).

### 3.5. Comparison of Arterial pH Levels Between HHFNC and nCPAP Groups

The comparison of the arterial pH between the nCPAP and HHFNC was carried out. Among patients receiving nCPAP, 59 (53.6%) had an arterial pH < 7.2, while 51 (46.4%) had a pH > 7.2. In the HHFNC group, 63 (57.3%) had a pH < 7.2, and 47 (42.7%) had a pH > 7.2. The overall distribution of arterial pH between the two groups was not statistically significant (*p* = 0.58). This suggests no significant difference in arterial pH outcomes between patients managed with nCPAP and those receiving HHFNC (Table 5).

## 4. Discussion

The nCPAP remains a primary respiratory support modality for neonates experiencing respiratory distress, offering significant benefits despite associated drawbacks such as nasal injury and the need for skilled nursing care [16]. Early initiation of nCPAP has been shown to reduce mechanical ventilation and surfactant administration, particularly in very preterm infants [17]. However, failure rates can reach 15–25%, prompting the exploration of alternative methods like HHFNC, which offers advantages such as reduced nasal trauma and improved ease of use [18]. In recent times, HHFNC has growing acceptance as an alternative respiratory support modality for preterm infants. HHFNC presents certain advantages over nCPAP, such as a reduced incidence of nasal trauma, utilization of patient and parent-friendly nasal prongs, and ease of use [19].

The present study aimed to compare two widely used respiratory support methods, HHFNC and nCPAP, in preterm neonates with respiratory failure. The mean age of preterm neonates was 14.72 ± 8.45 days for Group A and 15.07 ± 8.02 days for Group B. Previous studies reported lower mean ages of 1–2 days respectively, for preterm neonates with type 1 respiratory [20,21]. This considerable variation in mean age could potentially be attributed to differences in the age criteria used to include patients in the respective studies [22].

The gender distribution also showed no notable difference (*p* = 0.495), with a slightly higher percentage of females in both groups. Other studies reported higher male-to-female ratios of 1.66:1 reported by Sarkar et al. (2018) [23] and Sharma (2018) reported an even higher proportion of male preterm neonates (78.6%) in their study conducted in India [24]. The gestational age was similar between the two groups, indicating no significant differences. It is noteworthy that no previous study has been conducted specifically on preterm neonates in this particular comparison. However, studies conducted on preterm neonates have reported gestational ages of 32.5 ± 1.5 weeks by Shin et al. (2017) in Korea [25].

The mean SPO_2_ at admission suggests that the patients in this study presented with lower oxygen saturation levels, indicating severely compromised respiration. The higher mean SPO_2_ observed in the literature suggests better oxygenation levels in their study population, possibly reflecting more favourable respiratory status or different patient characteristics [26]. In contrast, the lower mean SPO_2_ reported in the prior study implies that their sample population had insufficient oxygenation levels at admission, perhaps indicating more serious respiratory distress [27]. Maintaining proper oxygen levels is critical for controlling hypoxia which can cause organ malfunction and consequences. Differences in mean SPO_2_ levels between studies might be attributed to patient characteristics, illness severity, or treatment procedures. Consistency in respiratory rate with previous research improves outcomes dependability and generalizability. However, increased respiratory rates may suggest changes in illness severity, patient characteristics, or clinical care strategies [28]. Similarly, the heart rate data from this investigation and the current literature add to the results’ consistency and dependability, as well as their generalizability to comparable patient groups. Previous reports of increased heart rates may imply variations in illness severity, patient characteristics, or clinical care techniques in their research cohorts [29].

Arterial pH levels did not differ significantly between the groups (*p* = 0.58), indicating similar efficacy in maintaining acid-base balance. This suggests that while nCPAP may improve survival, its impact on pH regulation is comparable to HHFNC. The arterial pH findings reported in this study and reported in other studies highlight consistency in the acid-base status of patients with similar respiratory conditions in different research settings. Acid-base disturbances are crucial indicators of the patient’s overall physiological status and can help identify the underlying cause and guide appropriate management. The similarity in PO_2_ values suggests comparable oxygenation levels in patients from both cohorts [30].

In this study, the mean duration of non-invasive ventilatory support and duration of hospital stay was significantly shorter in Group B compared to Group A. These results suggest that HHFNC may be associated with a shorter duration of both non-invasive ventilatory support and hospital stay compared to nCPAP. These findings align with results reported previously where the mean duration of application was also higher in the nCPAP group than the HHFNC group (69 ± 94.8 vs. 65 ± 99.9 h), but the difference was not statistically significant [10]. However, it is important to note that other studies have reported different findings. Some studies showed an inverse relation, where the mean duration of application was less in the nCPAP group than in the HHFNC group, but again the difference was not significant [31].

The longer mean duration of application in the nCPAP group in this study suggests that nCPAP may be utilized for a more extended period compared to HHFNC for providing non-invasive ventilatory support in this particular patient population. The variations in the duration of application between different studies may be influenced by differences in patient selection criteria, disease severity, and clinical management approaches. The lack of statistical significance in some studies could be due to the relatively small sample sizes or other factors that contribute to the variability in results.

Our results revealed no statistically significant differences in the rate of discharge without intubation (90.9% for nCPAP vs. 85.5% for HHFNC, *p* = 0.224) or mortality rates (4.5% vs. 3.6%). However, the intubation rate was higher in the HHFNC group (10.9% vs. 4.5%), though not statistically significant. These results suggest that while both modalities are effective, nCPAP may offer a slight advantage in reducing the need for intubation [32].

The Kaplan–Meier survival analysis demonstrated a significant survival benefit with nCPAP, with a longer mean (30.58 vs. 22.76 days) and median (33 vs. 24 days) survival time (*p* < 0.001). Cox regression analysis further supported this, showing a significantly lower risk of mortality or treatment failure with nCPAP (HR = 0.217, *p* < 0.001) [33]. Sepsis independently increased the risk of mortality (HR = 0.493, *p* = 0.030), highlighting its critical impact on neonatal outcomes. The higher incidence of treatment failure in the HHFNC group (compared to nCPAP) is noteworthy [34]. As reported in the literature indicates that HHFNC may be less effective in cases of severe respiratory distress, necessitating intubation or escalation of care [35,36]. This underscores the importance of patient selection when deciding between HHFNC and nCPAP as initial interventions. Studies advocate for clear clinical criteria to minimize the risk of treatment failure in vulnerable preterm neonates [37].

This study adds to the expanding body of evidence comparing HHFNC and nCPAP for treating respiratory insufficiency in preterm babies. While HHFNC has benefits such as shorter hospital stays and shorter ventilatory support duration, it may also be linked with increased rates of treatment failure and particular side effects such as nasal mucosal damage. These findings highlight the necessity of individualised therapy strategies based on clinical severity and patient-specific characteristics in optimising outcomes. The study’s applicability to other healthcare settings may be restricted, and it may not address long-term effects such as chronic respiratory diseases. Future multicentre research, including bigger, varied populations and standardised techniques, are required.

## 5. Conclusions

The duration of non-invasive ventilator support, shown by HHFNC, was significantly less when compared to nCPAP. Moreover, the length of hospital stay was also significantly shorter in the HHFNC group than in nCPAP. This suggests that HHFNC might be preferred for providing extended respiratory support in this patient population. Nonetheless, both HHFNC and nCPAP appear safe and effective in supporting preterm neonates through their respiratory challenges. Ultimately, individual patient needs and clinical considerations should guide the selection of the most suitable respiratory support method. Further research is needed to confirm these findings and explore potential strategies for mitigating adverse effects associated with HHFNC.

## Figures and Tables

**Figure 1 biomedicines-13-00602-f001:**
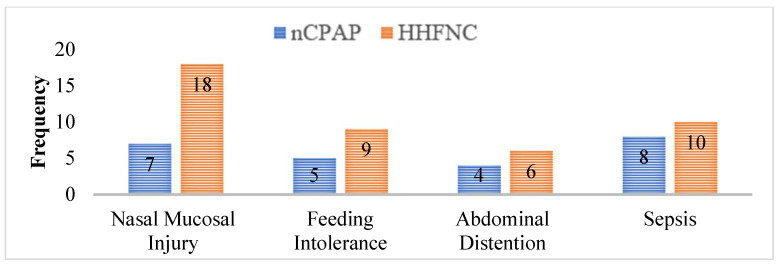
Comparison of adverse effects in both groups.

**Figure 2 biomedicines-13-00602-f002:**
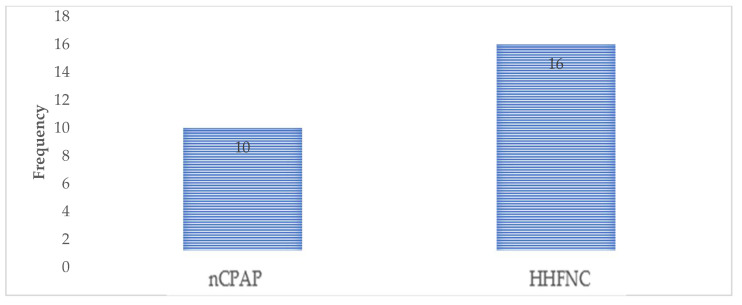
Comparison of treatment failure in both groups.

**Figure 3 biomedicines-13-00602-f003:**
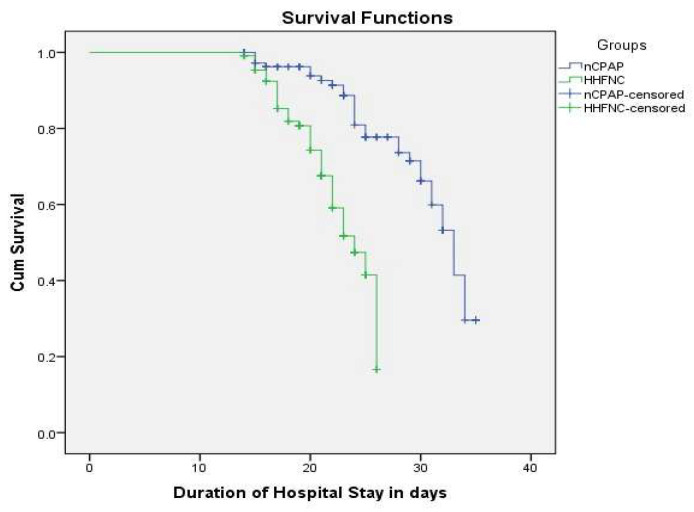
Kaplan–Meier survival analysis between the nCPAP and HHFNC groups.

**Table 1 biomedicines-13-00602-t001:** Comparison between the Groups at Baseline.

Characteristics	Group A (*n* = 128)	Group B (*n* = 128)	*p*-Value
Age (1–28 days)	14.72 ± 8.45	15.07 ± 8.02	0.796 ^a^
Gender			
Male	44 (40.0%)	49 (44.5%)	0.495 ^b^
Female	66 (60.0%)	61 (55.5%)	
Gestational Age (week)	32.71 ± 2.52	32.45 ± 2.94	0.472 ^a^
APGAR Score at 5 min	5.98 + 0.88	5.46 + 0.81	0.971 ^a^
SPO_2_ (%)	69.69 ± 18.2	70.43 ± 17.38	0.735 ^a^
Respiratory Rate (bpm)	52.29 ± 11.23	55.75 ± 11.66	0.028 ^a^
HR Rate (bpm)	134.28 ± 18.69	133.46 ± 17/83	0.765 ^a^
Arterial pH	7.26 ± 1.37	7.09 ± 1.41	0.301 ^a^
PCO_2_ (mmHg)	63.31 ± 12.1	63.02 ± 11.5	0.941 ^a^
PO_2_ (mmHg)	40.56 ± 5.42	39.87 ± 5.23	0.344 ^a^

^a^ Mann-Whitney U test. ^b^ Chi-Square test. Taking *p*-value ≤ 0.05 as significant.

**Table 2 biomedicines-13-00602-t002:** Comparison of Duration of Application of Non-invasive Ventilatory Support (hours) and hospital stay between the Groups.

Duration of Application of Non-Invasive Ventilatory Support (Hours)			
Study Group	Mean	Std. Deviation	*p*-Value
nCPAP (A)	72.29	20.7	0.013 ^a^
HHFNC (B)	65.20	15.9	
Hospital stays in days			
nCPAP (A)	24.25	6.07	0.000 ^a^
HHFNC (B)	20.14	3.50	

^a^ Mann-Whitney test, Taking *p*-value ≤ 0.05 as significant.

**Table 3 biomedicines-13-00602-t003:** Comparison of Duration of Neonatal Outcome between the Groups.

Neonatal Outcome	Groups		Total	*p*-Value
	nCPAP	HHFNC		
Discharged without the need for intubation	100 (90.9%)	94 (85.5%)	194 (88.2%)	^a^ 0.224
Expired	5 (4.5%)	4 (3.6%)	9 (4.1%)	
Intubated	5 (4.5%)	12 (10.9%)	17 (7.7%)	
Total	110	110	220	

^a^ Chi-Square test. Taking *p*-value ≤ 0.05 as significant.

**Table 4 biomedicines-13-00602-t004:** Cox Regression Analysis for Survival Outcomes.

Variable	B	SE	Wald	*p*-Value	Hazard Ratio (Exp B)	95% CI for Exp (B)
Treatment Group (nCPAP vs. HHFNC)	−1.526	0.295	26.806	0.000	0.217	(0.123–0.382)
Sepsis	−0.707	0.326	4.697	0.03	0.493	(0.257–0.947)
Treatment Failure	0.120	0.326	0.136	0.71	1.128	(0.588–2.163)

**Table 5 biomedicines-13-00602-t005:** Comparison of Arterial pH between the Groups.

Arterial pH	Groups		Total	*p*-Value
	nCPAP	HHFNC		
<7.2	59 (53.6%)	63 (57.3%)	122 (55.5%)	^a^ 0.58
>7.2	51 (46.4%)	47 (42.7%)	98 (44.5%)	
Total	110	110	220	

^a^ Chi-Square test. Taking *p*-value ≤ 0.05 as significant.

## Data Availability

The datasets generated for this study are available in the attachment.

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
