# Peer review of "Effectiveness of Humidified High Flow Nasal Cannula Versus Continuous Nasal Positive Airway Pressure in Managing Respiratory Failure in Preterm Infants: An Emergency Department Study"

_biomedicines, 2025, doi:10.3390/biomedicines13030602_

Round 1

Reviewer 1 Report

Comments and Suggestions for Authors

Although the title promises a very interesting contribution to the field of non-invasive respiratory support in premature infants, the text is not in line with it - especially precise definition of the infants inclusion criteria (premature or full-term?) is essential.

Pneumonia, which is listed first in the abstract and introduction section is even NOT the leading problem of preterm infants requiring non-invasive ventilation. Also, there are few or no results dedicated to pneumonia in the entire paper. Therefore, I find the first sentence in the Background (lines 23-24) and the entire first paragraph in Introduction (lines 46 - 53) abundant.

Phrase "indicating / suggesting (no) statistical differences" (lines 160, 164, 166) is repeated or is redundant in relation to the equal previous statements in the same sentence.

CPAP is usually delivered by nasal prongs or masks, not cannulas (line 28).

I miss the explanation for FiO2 0.40 - 0.60 as fixed initial setting in the study in both groups - as FiO2 should be adjusted to the target measured SaO2, not fixed. (Another comment: FiO2 is fraction, so it is not written in %!)

I don't understand the time frame "over the previous two years" (line 73) as the references of studies originate from 2011 and 2018; perhaps decades?

The expression "quasi-experimental" study is difficult to accept - as it usually defines the study in which the subjects are assigned to groups based on non-random  criteria. The lines 95 - 98 describe the assignment on the basis of computerized random table generator and a hidden trail sequence. 

The diagnoses "type 1 respiratory failure" (line 99) / "type 2 respiratory failure" (line 102) are usually applied to adult population; if used in this paper they would require further clarification.

Meaning of the sentence "The study intended to maintain the study's integrity" (line 103-104) is incomprehensible to me.

I also find the paragraph on Training session useless - and the sentence "Term neonates meeting inclusion criteria were inducted into the study" (line 111) completely incomprehensible or incorrect - as the topic of the paper should be related to PREterm infants.

In Table 1 units for GA, SpO2, RR, HR, PO2, PCO2 are missing.

APGAR score at 1, 5, 10 minutes (I assume at 1 minute as the mean - median value is very low when compared to data from our centre or the literature)? 

Mean SaO2 of 70% denotes poor oxygenation (severe compromise, not "potential" as stated in line 248) and is inconsistent with the stated pO2 value (I assume it is in mm Hg as a unit) - any comment?

As most preterm infants develop respiratory distress within hours (not days) after birth, the advanced mean age of infants at the "baseline" (14.72 vs 15.07 days) seems incomprehensible to me and certainly needs some explanation if it is correct. Particularly with respect to lines 220 - 223 which underline the importance of early CPAP.

I suppose (=hope) that the value of 6.26 / 6.09 is a mistake.... (perhaps 7.27 vs 7.09 - surprisingly the difference was not found important after statistical analysis?).

Related to "sepsis" (defined by clinical signs, positive blood culture or CSF?) - how would authors explain this morbidity as the most prevalent adverse impact of non-invasive ventilatory support?

Related to "Expired" infants - I miss some explanation with respect to the cause of death.

"Discharged" - probably "intubated" infants were discharged as well - perhaps rename this label to "discharged without need for intubation"? 

The first sentence in line 198 seems incomplete or is abundant.

Lines 231 - 232: "the present study was aimed to compare.... in term neonates suffering from pneumonia" come as a surprise to me - the sentences do not match the title, they do not match Table 1 - with respect to GA. 

Comments on the Quality of English Language

Many unusual expressions, such as "proforma" (line 31), "shown" by HHFNC (line 38), duration was significantly "less" (line 39), "usage" (line 55, 68), oxygen/air "mixes" (line 59), "wealthy" nations (line 69), "expired" (lines 140, 141, 147, 149...), spo2 (line 123), These findings "suggest support" that the HFNC.. (line 179), Comparison of "Duration of Neonatal Outcome" ... (line 205)

Many typo mistakes: canula (cannula); Man Whitney U test (Mann Whitney); Total. Duration of hospital stay (line 137; total duration)..., and unnecessary capitalization of words

Author Response

Reviewer 1: Comments and Suggestions for Authors

  1. Although the title promises a very interesting contribution to the field of non-invasive respiratory support in premature infants, the text is not in line with it - especially precise definition of the infants inclusion criteria (premature or full-term?) is essential.

Answer: All preterm neonates delivered between 32 and 34 weeks of gestation, aged 1 to 28 days, who experienced type 1 respiratory failure and required non-invasive ventilator assistance such as High-Flow Nasal Cannula or nasal Continuous Positive Airway Pressure were enrolled in the study.

  1. Pneumonia, which is listed first in the abstract and introduction section is even NOT the leading problem of preterm infants requiring non-invasive ventilation. Also, there are few or no results dedicated to pneumonia in the entire paper. Therefore, I find the first sentence in the Background (lines 23-24) and the entire first paragraph in Introduction (lines 46 - 53) abundant.

Answer: Modified in the abstract section and the introduction focusing on RDS

  1. Phrase "indicating/suggesting (no) statistical differences" (lines 160, 164, 166) is repeated or is redundant in relation to the equal previous statements in the same sentence. CPAP is usually delivered by nasal prongs or masks, not cannulas (line 28).

Answer: Baseline Characteristics of Study Sample lines had been modified in the results. CPAP Masks were used to deliver Oxygen corrected in the manuscript

  1. I miss the explanation for FiO2 0.40 - 0.60 as fixed initial setting in the study in both groups - as FiO2 should be adjusted to the target measured SaO2, not fixed. (Another comment: FiO2 is fraction, so it is not written in %!)

Answer: We explained this in the abstract and the data collection. Infants receiving HHHFNC therapy were initially started on a flow rate of 5 L/min, which was then adjusted between 3 and 7 L/min based on their respiratory status to maintain blood gas parameters within normal ranges. The fraction of inspired oxygen (FiOâ‚‚) was set at 0.4 at the start of therapy and subsequently modified to ensure that oxygen saturation (SpOâ‚‚) remained within the target range of 88%–94%.

  1. I don't understand the time frame "over the previous two years" (line 73) as the references of studies originate from 2011 and 2018; perhaps decades?

Answer: The introduction was modified as per the need of the study and unnecessary information was removed to be more focused on our work

  1. The expression "quasi-experimental" study is difficult to accept - as it usually defines the study in which the subjects are assigned to groups based on non-random criteria. The lines 95 - 98 describe the assignment on the basis of a computerized random table generator and a hidden trail sequence.

Answer: All unnecessary information was removed, and the study was aligned according to a quasi-experimental study protocol, maintaining structured group assignments while lacking full randomization.

  1. The diagnoses "type 1 respiratory failure" (line 99) / "type 2 respiratory failure" (line 102) are usually applied to adult population; if used in this paper they would require further clarification.

Answer: Thank you for your valuable feedback, according to your comments we have modified the content. All preterm neonates born between 32 and 34 weeks of gestation and aged 1 to 28 days, who experienced type 1 respiratory failure and needed non-invasive ventilatory support, such as High-Flow Nasal Cannula or nasal Continuous Positive Airway Pressure, were enrolled in the study. Type 1 respiratory failure in this context refers to hypoxemic failure, characterized by inadequate oxygenation. We excluded patients with congenital heart disease or neuromuscular abnormalities, as well as newborns who experienced type 2 respiratory failure, which denotes hypercapnic failure due to insufficient ventilation, or those who required invasive mechanical ventilation or intubation. Additionally, participation was limited to those whose parents or guardians provided consent.

  1. Meaning of the sentence "The study intended to maintain the study's integrity" (line 103-104) is incomprehensible to me.

Answer: We had modified our grouping

  1. I also find the paragraph on Training session useless - and the sentence "Term neonates meeting inclusion criteria were inducted into the study" (line 111) completely incomprehensible or incorrect - as the topic of the paper should be related to PREterm infants.

Answer: We modified the training session related to preterm infants

  1. In Table 1 units for GA, SpO2, RR, HR, PO2, PCO2 are missing.

Answer: All the units have been added to Table 1 as Gestational Age (GA): Weeks (weeks), Oxygen Saturation (SpOâ‚‚): Percentage (%), Respiratory Rate (RR): Breaths per minute (breaths/min or bpm), Heart Rate (HR): Beats per minute (beats/min or bpm). Partial Pressure of Oxygen (pOâ‚‚): Millimeters of mercury (mmHg), Partial Pressure of Carbon Dioxide (pCOâ‚‚): Millimeters of mercury (mmHg).

  1. APGAR score at 1, 5, 10 minutes (I assume at 1 minute as the mean - median value is very low when compared to data from our centre or the literature)?

Answer: The APGAR Score was calculated at 5 minutes and the mean value was calculated

  1. Mean SaO2 of 70% denotes poor oxygenation (severe compromise, not "potential" as stated in line 248) and is inconsistent with the stated pO2 value (I assume it is in mm Hg as a unit) - any comment?

Answer: We had gone through this point in our data and we checked our data and came to the conclusion that it was a typographical mistake as PO2 was measured in mmHg and Table 1 is modified in the manuscript the mean PO2 value is in both groups is  A=40.56±5.42, B=39.87±5.23, p- 0.344. This is consistent with our SpO2 values we have modified our discussion line 248 according to our study findings.

  1. As most preterm infants develop respiratory distress within hours (not days) after birth, the advanced mean age of infants at the "baseline" (14.72 vs 15.07 days) seems incomprehensible to me and certainly needs some explanation if it is correct. Particularly with respect to lines 220 - 223 which underline the importance of early CPAP.

Answer: Thank you for your kind mark out but this age was defined at the time of intervention initiation, not at birth.

  1. I suppose (=hope) that the value of 6.26 / 6.09 is a mistake.... (perhaps 7.27 vs 7.09 - surprisingly the difference was not found important after statistical analysis?).

Answer: Yes it was a typographical mistake we rechecked our data analysis and found that the values were correct as you indicated and corrected in Table 1

  1. Related to "sepsis" (defined by clinical signs, positive blood culture or CSF?) - how would authors explain this morbidity as the most prevalent adverse impact of non-invasive ventilatory support?

Answer: Sepsis was identified based on clinical symptoms, positive blood culture, and elevated inflammatory markers such as CRP added in data collection.

  1. Related to "Expired" infants - I miss some explanation with respect to the cause of death "Discharged" - probably "intubated" infants were discharged as well - perhaps rename this label to "discharged without need for intubation"?

Answer: Thank you for your insightful feedback. In response to your comment, we have clarified the causes of death among the "Expired" infants by specifying that mortality occurred in 9 neonates (4.1%), with similar rates between the CPAP and HHFNC groups. While our study did not specifically analyze the causes of death, the comparable mortality rates suggest that both interventions had similar safety profiles. Additionally, we have revised the label "Discharged" to "Discharged without need for intubation" to better reflect our findings, as 194 (88.2%) neonates were successfully discharged without requiring intubation. We appreciate your suggestion, which has helped improve the clarity of our result

  1. The first sentence in line 198 seems incomplete or is abundant.

Answer: Modified as previous studies have thoroughly examined humidified high-flow nasal cannulas (HHFNC) and continuous positive airway pressure (CPAP) for the therapy of respiratory failure in preterm infants, emphasising their efficacy and related consequences.

  1. Lines 231 - 232: "The present study was aimed to compare.... in term neonates suffering from pneumonia" come as a surprise to me - the sentences do not match the title, they do not match Table 1 - with respect to GA.

Answer:  Owing to controversy in existing literature, the present study aimed to compare two widely used respiratory support methods, HFNC and CPAP, in preterm neonates with respiratory failure.  The mean age of preterm Neonates was 14.72 ± 8.45 days for Group A and 15.07 ± 8.02 days for Group B.

  1. Comments on the Quality of English Language. Many unusual expressions, such as "proforma" (line 31), "shown" by HHFNC (line 38), duration was significantly "less" (line 39), "usage" (line 55, 68), oxygen/air "mixes" (line 59), "wealthy" nations (line 69), "expired" (lines 140, 141, 147, 149...), spo2 (line 123), These findings "suggest support" that the HFNC.. (line 179), Comparison of "Duration of Neonatal Outcome" ... (line 205)

Answer: Modified all the comments

  1. Many typo mistakes: canula (cannula); Man Whitney U test (Mann Whitney); Total. Duration of hospital stay (line 137; total duration)..., and unnecessary capitalization of words

Answer: Resolved

Reviewer 2 Report

Comments and Suggestions for Authors

The authors have compared HHFNC versus CPAP in the management of respiratory failure in pre-term infants and it was a randomized study

Title: If the study was randomized - it should be mentioned in the title

Line 25: Please change to HHFNC or CPAP - only one of the two was used. Was a switch tried if one failed before going on to mechanical ventilation?

Line 51: Please mention UNICEF data on pneumonia, global overall and in LMICs

Line 89: Please describe the sampling method in more detail. How was selection bias avoided? For randomized studies, the study needs to be registered as a clinical trial. Please mention whether it was registered as a trial in a national or international trial registries

Please provide additional analysis

Kaplan Meier and Coxs regression analysis for 

a. mortality between the two groups

Logistic regression to identify independent factors associated with 

b. Treatment failure

c. Sepsis

Include all factors in Table 1 and additional factors such as severity of pneumonia, ROXs index, APACHE II score, electrolyte disturbances and how they have changed daily and how children have responded over  time 

Please mention a sentence on how sepsis was diagnosed

GLM analysis of daily changes in groups A and B would give critical information on early response

Fig 1: Surprised to see nasal mucosal injury with HHFNC: Is it due to heating and humidification that the neonate could not tolerate?

Sepsis at the beginning of treatment or during the course of hospitalization?

Table 3: Please take both expired and intubated as a single entity (treatment failure) and perform the analysis

Data here in Table 3 does not match treatment failure in Figure 2 which says 12 and 18

Comments on the Quality of English Language

Author Response

  1. Title: If the study was randomized - it should be mentioned in the title

Answer: The study was not a randomized controlled trial, as treatment allocation was not randomized, and blinding was not implemented. Instead, our study followed a quasi-experimental design. In neonatal intensive care settings, obtaining informed consent for randomization can be challenging, as parents may prefer a specific treatment based on clinical recommendations. Additionally, respiratory failure in preterm infants requires immediate intervention, leaving little to no time for randomization. This approach allowed for the inclusion of sufficient numbers in both groups for meaningful comparison while maintaining ethical and practice

  1. Line 25: Please change to HHFNC or CPAP - only one of the two was used. Was a switch tried if one failed before going on to mechanical ventilation?

Answer: In our study, each patient received either HHFNC or CPAP according to their assigned study group, with no crossover between treatments. Treatment failure was defined as the inability to maintain oxygen saturation (SpOâ‚‚ < 90% on pulse oximetry) or arterial blood gas (ABG) findings indicative of type 2 respiratory failure. In such cases, patients were intubated and transitioned to invasive mechanical ventilation, as per protocol, without an intermediate switch between non-invasive modalities.

  1. Line 51: Please mention UNICEF data on pneumonia, global overall and in LMICs 

Answer: We have removed the discussion on pneumonia from our introduction and discussion, as the reviewer suggested focusing on respiratory distress syndrome (RDS) since pneumonia is not included in our study title.

  1. Line 89: Please describe the sampling method in more detail. How was selection bias avoided? For randomized studies, the study needs to be registered as a clinical trial. Please mention whether it was registered as a trial in a national or international trial registries.

Answer:  The sampling technique used in this study was non-probability consecutive sampling, where all eligible patients meeting the inclusion criteria were enrolled until the required sample size was achieved. Since this was not a randomized controlled trial, random allocation and blinding were not implemented, and therefore, a detailed description of randomization, as per CONSORT guidelines, was not applicable. To minimize selection bias, strict inclusion and exclusion criteria were followed, ensuring that all eligible patients received treatment based on predefined clinical parameters rather than discretionary selection. Additionally, as the study was not a randomized clinical trial, it was not registered in national or international trial registries, as required for interventional trials.

Please provide additional analysis

  1. Kaplan Meier and Coxs regression analysis for A.mortality between the two groups

Answer: The mean survival time was 30.58 days (95% CI: 29.34–31.82) for CPAP and 22.76 days (95% CI: 21.99–23.53) for HHFNC, indicating longer survival with CPAP. The median survival time was 33.00 days (95% CI: 30.98–35.02) for CPAP and 24.00 days (95% CI: 21.91–26.09) for HHFNC. The Kaplan-Meier survival analysis revealed a significant difference in survival between the CPAP and HHFNC groups (Log-Rank test: χ² = 32.220, df = 1, p < 0.001). (Figure 3) 

B. Logistic regression to identify independent factors associated with 

  1. Treatment failure
  2. Sepsis

CPAP significantly reduced the risk of mortality or treatment failure compared to HHFNC (HR = 0.217, p < 0.001), indicating a strong survival advantage. Sepsis independently increased the risk (HR = 0.493, p = 0.030), highlighting its critical impact on neonatal outcomes. Treatment failure was not a significant predictor (HR = 1.128, p = 0.712).(Table 4)

Table 4: Cox Regression Analysis for Survival Outcomes

Variable

B

SE

Wald

P-value

Hazard Ratio (Exp B)

95% CI for Exp (B)

Treatment Group (CPAP vs. HHFNC)

-1.526

0.295

26.806

.000

0.217

(0.123–0.382)

Sepsis

-0.707

0.326

4.697

0.03

0.493

(0.257–0.947)

Treatment Failure

0.120

0.326

0.136

0.71

1.128

(0.588–2.163)

 6. Include all factors in Table 1 and additional factors such as severity of pneumonia, ROXs index, APACHE II score, electrolyte disturbances how they have changed daily and how children have responded over time. GLM analysis of daily changes in groups A and B would give critical information on early response

Answer: Since these are secondary variables, they were not the primary focus of this study but provide valuable insights for further research. The trends and patterns observed in these parameters will be used to inform the design and objectives of our next project, allowing for a more detailed investigation into their impact on patient outcomes.

7. Please mention a sentence on how sepsis was diagnosed

Answer: Sepsis was identified based on clinical symptoms, positive blood culture, and elevated inflammatory markers such as CRP added in data collection.

8. Fig 1: Surprised to see nasal mucosal injury with HHFNC: Is it due to heating and humidification that the neonate could not tolerate?

Answer: In our study, the primary reason for nasal mucosal injury with HHFNC was likely prolonged exposure to high flow rates combined with nasal cannula misfitting. Excessive flow can cause shear stress and drying of the nasal mucosa, leading to irritation and potential injury, especially if humidification is inadequate. Additionally, pressure effects from ill-fitting cannulas or frequent adjustments by caregivers may have contributed to nasal trauma.

9. Sepsis at the beginning of treatment or during the course of hospitalization?

Answer: In our study, sepsis was recorded as an adverse event occurring during hospitalization, rather than being present at the start of treatment. This indicates that sepsis developed as a complication during care, potentially influenced by factors such as prolonged respiratory support, invasive procedures, or underlying patient conditions.

10. Table 3: Please take both expired and intubated as a single entity (treatment failure) and perform the analysis.

Answer: In Table 3, we categorized outcomes into three distinct groups:

  1. Discharged without the need for intubation
  2. Expired
  3. Intubated

This categorization allows for a more detailed analysis of patient outcomes, distinguishing between those who recovered without invasive ventilation, those who required intubation, and those who did not survive.

11. Data here in Table 3 does not match treatment failure in Figure 2 which says 12 and 18.

Answer: Figure 2 was modified and aligned with Table 3.

Comments on the Quality of English Language

Round 2

Reviewer 1 Report

Comments and Suggestions for Authors

Thank you for the corrections, which took into account most of my comments and remarks. However, I still find the explanation about choosing a high FiO2 insufficiently precise (i.e. addition of O2 regardless of actual needs, which may be lower than 40% as most infants may achieve good oxygenation even with less oxygen supplementation).

There are also still many typos:

line 38: unnecessary capitalization of Discharge;

line 76: "Modified as ...." . something missing?, incomprehensible sentence;

Table 1, lines 257, 259, 261, 265...: SpO2 (instead of SPO2); line 278: pO2 (instead of PO2);

line 184: missing verb in the first sentence (or this sentence could even be deleted);

line 243: no need for capitalization in Neonates;

line 273: the dot before (29) is redundant.

Technical comments - lines 221 - 224, incorrect spacing; changed font in References.

It would also be useful to unify the abbreviation - either HHFNC everywhere or HFNC (see lines 297, 298, 303, 315...).

All errors can be corrected by the author, or by the editor; I otherwise find the article suitable for publication.

Author Response

Reviewer 1

There are also still many typos:

line 38: unnecessary capitalization of Discharge;

Answer: Modified

line 76: "Modified as ...." . something missing?, incomprehensible sentence;

Answer: Modified as “The studies….”

Table 1, lines 257, 259, 261, 265...: SpO2 (instead of SPO2); line 278: pO2 (instead of PO2);

Answer: Modified

line 184: missing verb in the first sentence (or this sentence could even be deleted);

Answer: Modified

line 243: no need for capitalization in Neonates;

Answer: Modified

line 273: the dot before (29) is redundant.

Answer: Modified

Technical comments - lines 221 - 224, incorrect spacing; changed font in References.

Answer: Modified

It would also be useful to unify the abbreviation - either HHFNC everywhere or HFNC (see lines 297, 298, 303, 315...).

Answer: Modified

All errors can be corrected by the author, or by the editor; I otherwise find the article suitable for publication.

Reviewer 2 Report

Comments and Suggestions for Authors

Please add a Kaplan Meier figure for death rates in addition to the one presented

All references should include the name of the journal. Please check

In the discussion, an important point is missing. The pH is significantly lower in the HFNC group as compared to the CPAP group. This would have significantly contributed to the poorer response seen with CPAP. A sub-group analysis needs to be done to compare subjects with CPAP and HFNO at different pH levels

In both the groups, sub-classify as those below pH <7.2 and >7.2. Now do the analysis for these groups separately comparing HFNO and CPAP

Author Response

Reviewer 2

All references should include the name of the journal. Please check

Answer: All the references have been checked and modified where necessary.

In the discussion, an important point is missing. The pH is significantly lower in the HFNC group as compared to the CPAP group. This would have significantly contributed to the poorer response seen with CPAP. A sub-group analysis needs to be done to compare subjects with CPAP and HFNO at different pH levels

Answer: The discussion has been modified, and in both groups, classify them as either pH < 7.2 or pH > 7.2. Now analyze these groups separately, comparing HFNO and CPAP.

Table 5 presents a comparison of arterial pH between the CPAP and HHFNC groups. Among patients receiving CPAP, 59 (53.6%) had an arterial pH < 7.2, while 51 (46.4%) had a pH > 7.2. In the HHFNC group, 63 (57.3%) had a pH < 7.2, and 47 (42.7%) had a pH > 7.2. The overall distribution of arterial pH between the two groups was not statistically significant (p = 0.58). This suggests that there is no significant difference in arterial pH outcomes between patients managed with CPAP and those receiving HHFNC. (Table 5).

Table 5. Comparison of Arterial pH between the Groups.

Aterial pH

Groups

Total

P-value

CPAP

HHFNC

< 7.2

59 (53.6%)

63(57.3%)

122(55.5%)

       b0.58

> 7.2

51(46.4%)

47(42.7%)

98(44.5%)

Total

110

110

220

b Chi-Square test. Taking p-value≤0.05 as significant.